# `NeuroEvoBench`: Benchmarking Evolutionary Optimizers for Deep Learning Applications

**Robert Tjarko Lange**[*]
Technical University Berlin
Science of Intelligence Cluster of Excellence

**Yujin Tang**
Google DeepMind

**Yingtao Tian**
Google DeepMind

## Abstract

Recently, the Deep Learning community has become interested in evolutionary optimization (EO) as a means to address hard optimization problems, e.g. meta-learning through long inner loop unrolls or optimizing non-differentiable operators. One core reason for this trend has been the recent innovation in hardware acceleration and compatible software – making distributed population evaluations much easier than before. Unlike for gradient descent-based methods though, there is a lack of hyperparameter understanding and best practices for EO – arguably due to severely less "graduate student descent" and benchmarking being performed for EO methods. Additionally, classical benchmarks from the evolutionary community provide few practical insights for Deep Learning applications. This poses challenges for newcomers to hardware-accelerated EO and hinders significant adoption. Hence, we establish a new benchmark of EO methods (`NeuroEvoBench`) tailored toward Deep Learning applications and exhaustively evaluate traditional and meta-learned EO. We investigate core scientific questions including resource allocation, fitness shaping, normalization, regularization & scalability of EO. The benchmark is open-sourced at `https://github.com/neuroevobench/neuroevobench` under Apache-2.0 license.

## 1 Introduction

The Deep Learning revolution has been largely enabled by the (arguably) "unreasonable effectiveness" of gradient descent (GD)-based optimization in high-dimensional search spaces of parameters (Kleinberg et al., 2018; Hardt et al., 2016; Ge et al., 2015). However, a plethora of challenging high-dimensional optimization problems where GD methods are inadequate exist, including not only hyperparameter search but also the optimization of non-differentiable operators (e.g. objective or architecture, Tian and Ha, 2022; He et al., 2021), the computation of ill-behaved gradients through long computational graphs (Peyré et al., 2017), and applications requiring black-box optimization (Metz et al., 2021). Such challenges have sparked a recent resurgence of interest in modern scalable EO methods by the machine learning community. Still, compared to widely-studied GD optimizers, there remains a lack of intuition for EO methods. We argue that this is due to three reasons:

**Hardware lottery phenomenon** (Hooker, 2021): Until recently neuroevolution experimentation required a significant amount of engineering overhead caused by the orchestration of parallelized population rollouts (e.g. using Dask or Ray, Moritz et al., 2018). This has obstructed replicability and hindered EO from leveraging the recent advances in hardware acceleration.

**Lack of 'graduate-student' descent** (Gencoglu et al., 2019): The last decade has seen a strong increase in GD optimization practitioners. This facilitated the discovery of robust hyperparameter basins that perform well on standard research tasks. EO, on the other hand, lacks behind in human

---

[*]Work was partially done while being at Google DeepMind. Contact: `robert.t.lange@tu-berlin.de`.

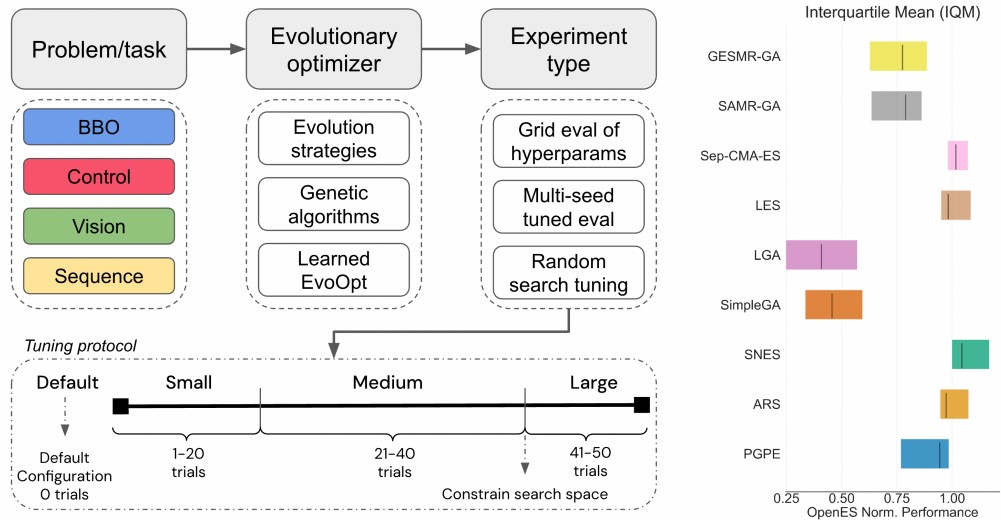

Figure 1: Proposed `NeuroEvoBench` benchmark. **Left.** The benchmark is composed of four classes of problems, different classes of EO and experiment types including a random search tuning protocol with different budgets (default, small, medium, large). **Right.** Aggregated normalized median performance across EO algorithms on nine neuroevolution tasks, sorted by their median performances and across 5 seeds. Genetic Algorithms are generally outperformed by Evolution Strategies.

capital investment and sufficient hyperparameter intuition. This has been partly caused by engineering challenges and the lack of adequate hardware acceleration and the required software.

**Missing ML-relevant EO benchmarks**: The majority of existing benchmarks focus on synthetic functions and standard black-box optimization tasks (e.g. BBOB, Hansen, 2009). But neural network fitness landscapes can fundamentally differ from these settings, limiting the meaningfulness of any drawn conclusions for relevant practical cases of neuroevolution. This is due to differences in the nature of considered problems (e.g. robot policy parameters) and the size of the search space.

However, recent advances have shifted the balance. For example, we have seen tremendous advances in both hardware acceleration (e.g. GPU/TPU) as well as general-purpose frameworks/infrastructures that facilitate smooth development of distributed programs (e.g. PyTorch and JAX, Bradbury et al., 2018). These developments enable not only data- and model-parallelism for GD-based training pipelines but also large-scale EO with auto-vectorized and device-parallel population evaluations, e.g. `EvoJAX` (Tang et al., 2022) and `evosax` (Lange, 2022a). This makes it the right time to reassess the uncertainty about EO use cases, problem-dependent EO choices, and optimal hyperparameter settings. We, therefore, introduce `NeuroEvoBench` (NEB), a benchmark for EO methods tailored to relevant Deep Learning applications and leveraging such advances in the hardware and software stack. This allows for effective scientific validation of new algorithmic improvements at low iteration times. As shown in Figure 1, our proposed benchmark contains four classes of different, selected black-box optimization tasks, each with three experiment types including a random search tuning protocol with different budgets. Using NEB, we can investigate several core scientific questions, such as the performance of different EO, and the impact of different decisions such as population size, and dimensionality, to give the right picture for future reference. Our contributions are summarized as:

1. We construct a hardware-accelerated EO benchmark providing relevant scientific insight for the ML community. It consists of 11 selected black-box optimization tasks and a standardized experiment protocol. We execute the benchmark for 10 EO methods including both Evolution Strategies (ES) and Genetic Algorithms (GA) and analyze their robustness.

2. We open-source the benchmark for a broader adoption, to provide evidence for future analysis and make benchmarking new EO method easier. Furthermore, we make a quick overview for reference available online: https://sites.google.com/view/neuroevobench.

3. We investigate several core scientific questions with regard to resource allocation, gradient optimizer choice and regularization techniques. We further investigate the scaling of EO with respect to parameters, population size and the number fitness evaluations per candidate.

## 2 Related Work & Background

**Gradient-Based Optimization Benchmarks.** The success of Deep Learning methods has arguably been enabled by the "unreasonable" success of GD-based optimization (Kleinberg et al., 2018; Hardt et al., 2016; Ge et al., 2015) and the adoption of standardized benchmark tasks (e.g. ImageNet, Deng et al., 2009). Furthermore, there exist several benchmarks for gradient-based descent optimization, e.g. Schaul et al. (2013); Schneider et al. (2019); Metz et al. (2020) which vary in the runtime and number of optimization tasks. However, these benchmarks consider only a subspace of imaginable solutions due relying on explicit well-behaved gradient evaluations and not natively supporting distributed population rollouts required for EO methods. EO, on the other hand, deals with problems with fewer restrictions, e.g., long-horizon tasks or non-differentiable dynamics.

**Black-Box Optimization Benchmarks.** For BBO methods, on the other hand, the benchmarks have largely focused on synthetic functions or problems, e.g. the BBOB (Hansen, 2009), HPO-B (Arango et al., 2021), nevergrad (Bennet et al., 2021) and Vizier (Golovin et al., 2017) benchmarks. This limits their relevance for large-scale BBO problems and Deep Learning-specific practitioners. Here we aim to overcome this limitation by introducing `NeuroEvoBench` which explicitly targets neuroevolution tasks of varying scale (parameter count, problem domain, resource constraints, etc.). Finally, Mousavirad et al. (2020) introduce a benchmark evaluating meta-heuristic algorithms on neural network tasks. Unfortunately, they do not consider ES/GA and the code is closed-source.

**Evolutionary Optimization.** EO constitute a set of random search algorithms based on the principles of biological evolution. In this benchmark, we focus on two classes of EO methods:

1. Evolution Strategies (ES): ES adapt a parameterized distribution (e.g. multivariate normal) to iteratively search for well performing solutions. After sampling a population of candidates, their fitness is estimated using Monte Carlo (MC) evaluations. The scores are used to update the search distribution in order to maximize the likelihood of well-performing candidates.

2. Genetic Algorithms (GA): Unlike ES, GA does not rely on a uni-modal search distribution. Instead, they keep an archive of 'parent' solutions from which new 'children' candidates are sampled and adapted via mutation. After evaluation, well-performing children are selected to replace parents. GAs differ in their replacement and mutation rate adaptation strategy.

**Accelerated Neuroevolution.** Traditionally, EO has required a substantial amount of software engineering to orchestrate parallel evaluations of fitnesses in polulation. We argue that this additional overhead has largely held back EO progress. `NeuroEvoBench` aims to leverage a set of software improvements that facilitate auto-vectorized and device parallel evaluations using the JAX (Bradbury et al., 2018) ecosystem. More specifically, we use accelerated fitness evaluations implemented by `EvoJAX` (Tang et al., 2022) and EO algorithms from `evosax` (Lange, 2022a). These advances in turn significantly reduce the benchmark runtime and allow for seamless GPU/TPU acceleration.

## 3 `NeuroEvoBench`: Problems, Optimizers, Fitnesses and Experiment Types

We introduce the `NeuroEvoBench` benchmark (see Figure 1) to provide insights into the performance of EO algorithms for Deep Learning problems. Our design is strongly based on the choices made by Schmidt et al. (2021, GD case). At its core a `NeuroEvoBench` experiment combines a problem with an Evolutionary Optimization algorithm and a fitness shaping operation:

$$
\underbrace{\begin{Bmatrix} \text{P1} \\ \text{P2} \\ \dots \\ \text{P11} \end{Bmatrix}}_{11} \times \underbrace{\begin{Bmatrix} \texttt{OpenAI-ES} \\ \texttt{PGPE} \\ \dots \\ \texttt{Sep-CMA-ES} \end{Bmatrix}}_{10} \times \underbrace{\begin{Bmatrix} \text{raw} \\ \text{z-score} \\ \text{ranks} \\ \text{[-1, 1] norm} \end{Bmatrix}}_{4} \times \underbrace{\begin{Bmatrix} \text{Random Search} \\ \text{Multi-Seed Eval} \\ \text{Grid search} \end{Bmatrix}}_{3}
$$

**Problems** **Optimizers** **Fitnesses** **Experiment Types**

More specifically, they include different optimization substrates, fitness functions, and optimization resource budgets, i.e. population sizes and number of stochastic evaluations.[2] We detail each component of `NeuroEvoBench` below (see also Table 1).

## 3.1 Problems

NEB implements a set of 11 problems from 4 core classes chosen to cover a wide range of settings. All of the tasks are implemented in JAX (Bradbury et al., 2018) and thereby can be directly evaluated on accelerators, speeding up the parallel population fitness computation significantly.

**BBO** . For comparability, we host a set of BBO benchmark tasks including the noiseless BBOB (Hansen, 2009) and the HPO-B (Arango et al., 2021, continuous) sweep. These are concerned with the optimization of analytical functions with different characteristics (multi-modal, conditioning, etc.) and surrogate functions extracted from hyperparameter settings of small machine learning models.

**Control** . We evaluate EO performance on a set of 4 control / Reinforcement Learning tasks including continuous control (Brax, Freeman et al., 2021) tasks using small Tanh MLP policies and MinAtar (Young and Tian, 2019; Lange, 2022b) visual control tasks using CNN for policies. The fitness score is computed as the cumulative episode return obtained by the parametrized policies.

**Vision** : We consider a set of vision classification and generation tasks including MNIST image generation (via a MLP VAE, Kingma and Welling, 2013), Fashion-MNIST (2 layer CNN), and CIFAR-10 (All-CNN-C variant) classification. The training fitness used the VAE and cross-entropy loss on the training set, while the evaluation used the test set and accuracy.

**Sequence** : We make use of two sequence prediction tasks: The addition regression (Le et al., 2015) task using GRU RNN and Sequential MNIST classification (LSTM, Hochreiter and Schmidhuber (1997)). Both tasks test the ability of EO in optimizing systems with exploding/vanishing gradients.

Table 1: Summary of problems used in our benchmark experiments. Note that problems 1 and 2 consider surrogate optimization tasks/models (Hansen, 2009; Arango et al., 2021), while all other tasks are concerned with the evolutionary optimization of neural network weights. Throughout the main text, we focus on such tasks but implement the BBO problems for completeness.

|        | Data     | Model     | Task      | Metric | Obj.   | Pop. | Gens. | Time | Dim.    |
|--------|----------|-----------|-----------|--------|--------|------|-------|------|---------|
| **P1** | HPO-B    | HPO       | Surrogate | Perf.  | Perf.  | 4    | 100   | 5m   | $< 20$  |
| **P2** | BBOB     | BBO       | Synthetic | Perf.  | Perf.  | 32   | 100   | 2m   | $< 50$  |
| **P3** | Ant      | MLP       | Control   | Return | Return | 512  | 2k    | 30m  | 4136    |
| **P4** | Fetch    | MLP       | Control   | Return | Return | 512  | 2k    | 30m  | 4650    |
| **P5** | Asterix  | CNN       | Control   | Return | Return | 256  | 1.5k  | 1h   | 51989   |
| **P6** | Breakout | CNN       | Control   | Return | Return | 256  | 1.5k  | 1h   | 51923   |
| **P7** | MNIST    | VAE       | Generate  | Loss   | Loss   | 256  | 10k   | 7m   | 52984   |
| **P8** | F-MNIST  | CNN       | Classify  | Acc.   | Loss   | 128  | 4k    | 5m   | 11274   |
| **P9** | CIFAR-10 | All-CNN-C | Classify  | Acc.   | Loss   | 128  | 2.5k  | 1h   | 3994    |
| **P10**| Addition | GRU       | Regress   | Loss   | Loss   | 128  | 5k    | 5m   | 3425    |
| **P11**| S-MNIST  | LSTM      | Classify  | Acc.   | Loss   | 512  | 3k    | 1h   | 10090   |

Our task selection is motivated by the observation that small-scale BBO benchmarks alone (e.g. BBOB/HPO-B) do not suffice in predicting the performance of EO methods on high-dimensional tasks requiring the optimization of network weights (see comparison of Figures 1 and 11). Furthermore, the different task classes cover a wide range of representative Deep Learning problems required for robust performance evaluation (see Figure 10).

## 3.2 Optimizers and Fitnesses

Concretely, `NeuroEvoBench` focuses on ten competitive EO selected from the vlasses of ES and GA that represent different trade-off between exploration and exploitation in parameter space (Table 2):

---

[2] The majority of considered tasks were chosen to be evaluated within less than 30 minutes (estimated on a single V100S GPU). This allows for rapid evaluation of parameter tuning in <1 day of sequential execution time.

Table 2: Summary of Evolutionary Optimizers used in our benchmark experiments.

| | ES1 | ES2 | ES3 | ES4 | ES5 |
|---|---|---|---|---|---|
| **EO Name** | OpenAI-ES | PGPE | ARS | SNES | Sep-CMA-ES |
| **Reference** | Salimans et al. (2017) | Sehnke et al. (2010) | Mania et al. (2018) | Schaul et al. (2011) | Ros and Hansen (2008) |
| **EO Type** | FD-GD | FD-GD | FD-GD | FD-GD | EoD |
| | **ES6** | **GA1** | **GA2** | **GA3** | **GA4** |
| **EO Name** | LES | Gaussian-GA | SAMR-GA | GESMR-GA | LGA |
| **Reference** | Lange et al. (2022) | Rechenberg (1978) | Clune et al. (2008) | Kumar et al. (2022) | Lange et al. (2022) |
| **EO Type** | Meta-EoD | GA | SA-GA | SA-GA | Meta-SA-GA |

Finite Difference-based ES: A subset of ES use random perturbations to MC estimate a finite difference gradient to the fitness function, $F(.)$:

$$\nabla_\theta \mathbb{E}_{\epsilon \sim \mathcal{N}(0,I)} F(\theta + \sigma\epsilon) = \frac{1}{\sigma} \mathbb{E}_{\epsilon \sim \mathcal{N}(0,I)}[F(\theta + \sigma\epsilon)\epsilon]$$

This estimate is then used along standard GD-based optimizers to refine the search distribution mean $\theta$. ES vary in their use of fitness transformation, anti-correlated noise, elite selection and covariance.
Estimation-of-Distribution ES: A second class of ES does not rely on noise perturbations or low-dimensional approximations to the fitness gradient. Instead, algorithms such as Sep-CMA-ES (Ros and Hansen, 2008) rely on elite-weighted mean updates and iterative covariance estimation.
Gaussian (Self-Adaptation) GA: We consider GAs that utilize Gaussian isotropic perturbations to the sampled children and do not use cross-over. We evaluate a simple GA with top-k elitist parent replacement (Rechenberg, 1978) as well as two GAs that adapt the mutation rates based on the parent performance (SAMR-GA, Clune et al. (2008) and GESMR-GA, Kumar et al. (2022)).
Learned Evolutionary Optimization: We study the performance of two meta-learned EO algorithms (Learned ES, Lange et al. (2022) and Learned GA, Lange et al. (2023)). Instead of relying on manually designed update rules, learned EO leverage self-attention to parameterize novel families of EO algorithms. The corresponding parameters are meta-evolved on a small set of BBO problems.

## 3.3 Experiment Types

**Random search with space refinement**. The core experiment of `NeuroEvoBench` is a random search tuning run with different budgets. More specifically following Schmidt et al. (2021), we consider four settings: Default parameter configuration, `small` - 20 trials, `medium` - 40 trials and `large` - using 50 trials with a search space refinement after 40 runs using the 10 best hyperparameter settings. We provide additional information on the tuning ranges in the appendix.

**Multi-seed re-evaluation of tuned hyperparameters**. After completing the random search tuning, we re-evaluate the best configuration on multiple random seeds. This tests the robustness of the tuning procedure to the stochasticity of the training run. Whenever applicable, we make use of the robust evaluation metrics introduced by the `rliable` library (Agarwal et al., 2021).

**Grid search evaluation of core settings**. Finally, we evaluate certain hyperparameter settings more explicitly by running a grid sweep. These include different fitness score transformations for the FD-GD ES methods, mean decay regularization, and resource settings (population/evaluation).

## 3.4 `NeuroEvoBench` Task Evaluation API

```python
from evosax import Strategies
from neuroevobench.problems.cifar import CifarPolicy
from neuroevobench.problems.cifar import CifarTask
from neuroevobench.problems.cifar import CifarEvaluator
# Define the task-specific network policy
policy = CifarPolicy()
# Instantiate the train and test task
train_task = CifarTask(train_batch_size, test=False)
test_task = CifarTask(test_batch_size, test=True)
```

```
# Instantiate the task evaluator and run the evo search loop
evaluator = CifarEvaluator(
    policy, train_task, test_task, popsize, strategy, ...)
evaluator.run(num_generations, eval_every_gen)
```

Listing 1: Example for `NeuroEvoBench` task evaluation API for CIFAR-10 Classification.

We provide an easy-to-use API for `NeuroEvoBench`. An example of CIFAR-10 (Krizhevsky et al., 2009) classification is shown in Listing 1. Our API expects three ingredients: The 'Policy' (neural network/agent architecture), the 'Task' (accelerated/distributed fitness evaluator) and the 'Evaluator' (the iterative training/generation loop). Plus, the EO algorithm only needs to follow the standard `ask-tell` API used for example in `evosax` (Lange, 2022a).

## 4 Results

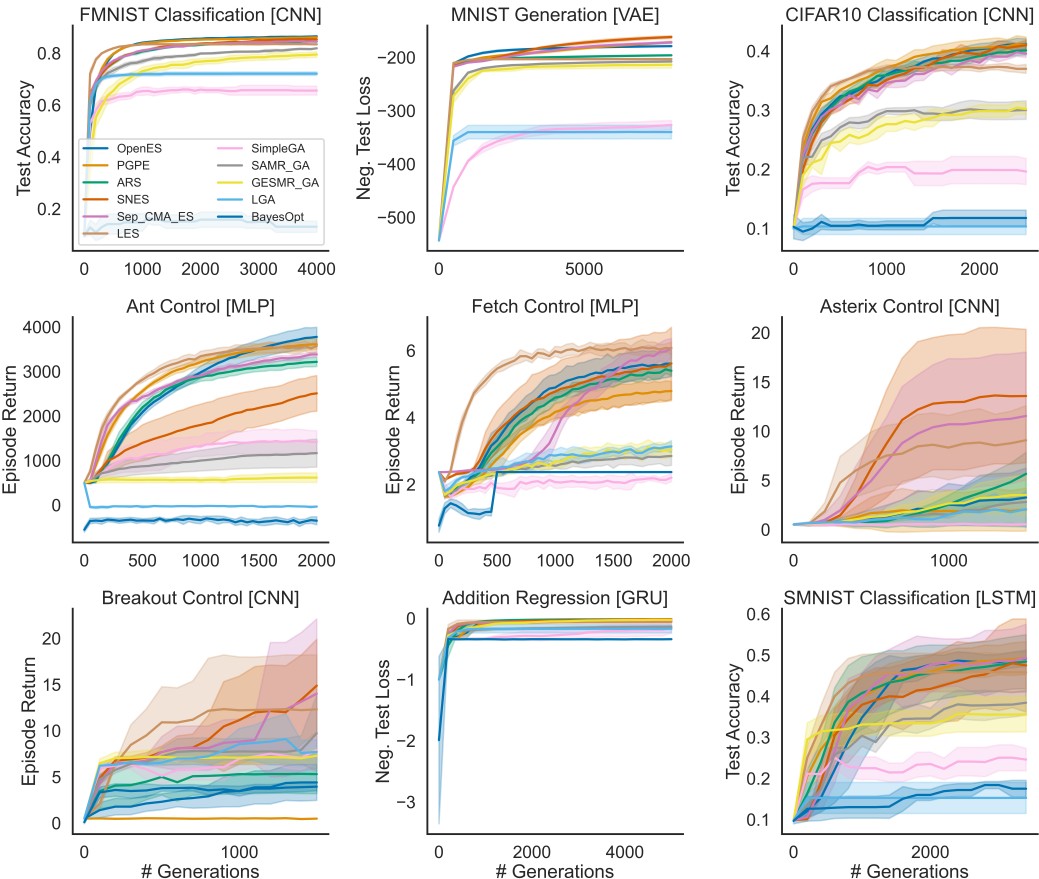

Figure 2: `NeuroEvoBench` task evaluation after 50 trials of random search. We plot the tuned performance of all 10 considered EO for the 9 neuroevolution tasks. GAs are generally outperformed by ES alternatives, but there is no clear winner across all considered ES. The results are averaged over 5 independent runs & we plot standard error bars.

We conduct a set of experiments with `NeuroEvoBench` and discuss the lessons learned. For this, we focus on the neuroevolution tasks (P3-P11) in order to provide general insights to the ML community.

### 4.1 Given a predefined tuning budget, which EO performs best?

Is there a single EO algorithm that dominates all others across neuroevolution tasks? In Figure 2 we plot the performance of the 10 considered EOs for the 9 neuroevolution tasks after random search

tuning (see Section 4.2 for more details). We find that there is no single clear winner (see also Figure 1 for the aggregated results) across all settings. Furthermore, we observe that the different Genetic Algorithm variants are generally outperformed by the ES competitors. Finally, while the meta-learned EO algorithms can outperform the manually designed alternatives on single tasks (e.g. Fetch and Ant control), they appear to be only capable of limited generalization beyond their meta-training setting.

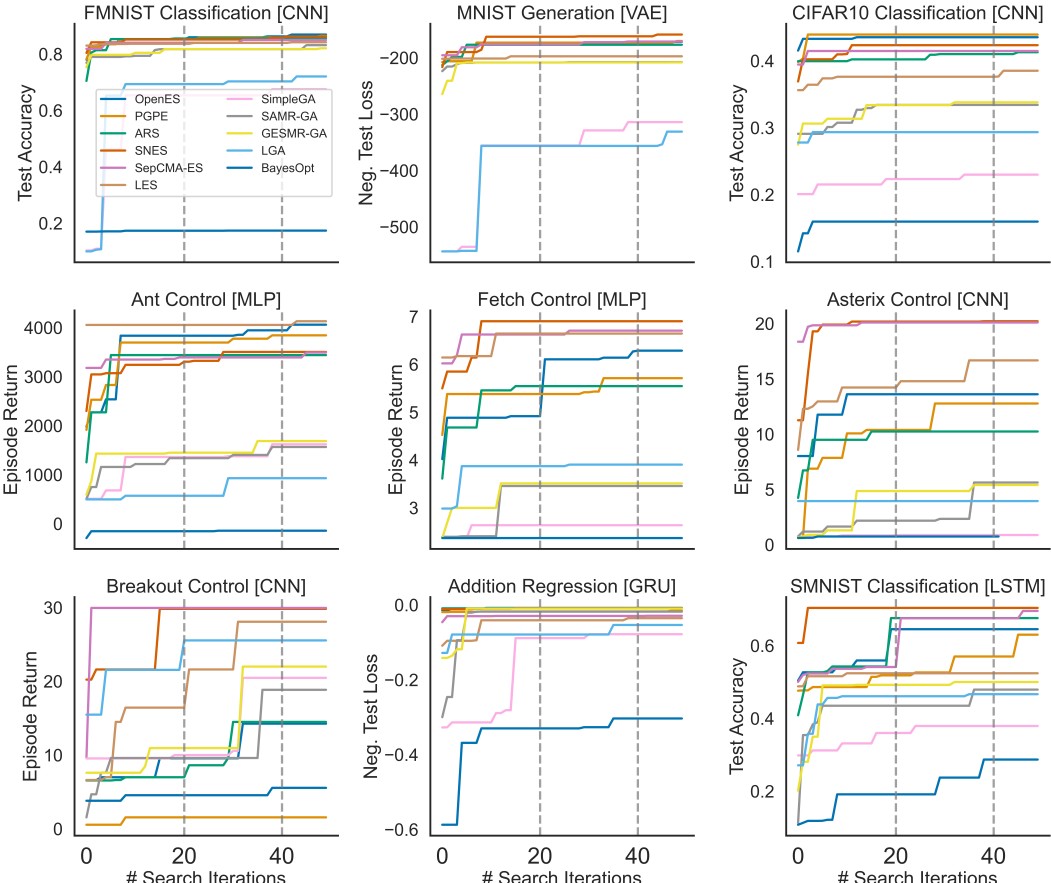

Figure 3: `NeuroEvoBench` best task performance across 50 trials of random search. We plot the maximum performance tracked across the random search procedure for all 10 considered EO and for the 9 neuroevolution benchmark tasks. We consider a single random run for the tuning process.

## 4.2 How much tuning is required to achieve competitive performance?

Next, we investigated the impact of the tuning budget on the performance of the EO. Ideally, EO do not require too much tuning in order to perform well across different classes of tasks. In Figure 3 we find that indeed most of the considered optimizers find their peak performance within the small tuning budget and less than 20 random search tuning trials. We note that for most tasks a single evaluation of the hyperparameters appears to provide a robust performance estimate. Only for the Asterix and Breakout tasks we observed a significant drop in the estimated performance when re-evaluating the best performance over multiple random seeds (see Figure 2).

## 4.3 What are general EO recommendations and their scaling properties?

We investigate a set of core considerations when performing Evolutionary Optimization: Resource allocation, the effect of stochastic evaluation fitness evaluation, fitness transformations and regularization techniques. More specifically for this exercise, we focus on OpenAI-ES (Salimans et al., 2017) as a popular representative of finite-difference ES. EO rely on stochastic fitness evaluations of the individual population members. This introduces an explicit choice of resource allocation: Shall

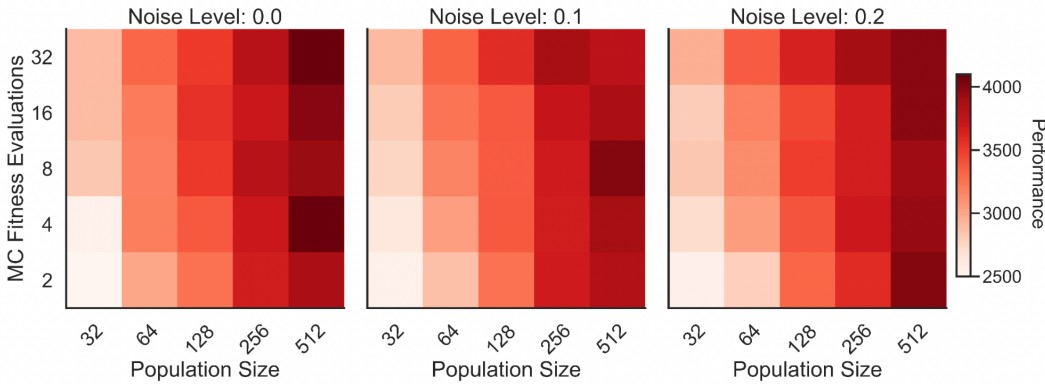

Figure 4: Resource allocation for EO. For a noisy Ant task we find that more candidate evaluations lead to better downstream performance but require more hardware memory. This is independent of the external noise level. The results are averaged over 3 independent runs.

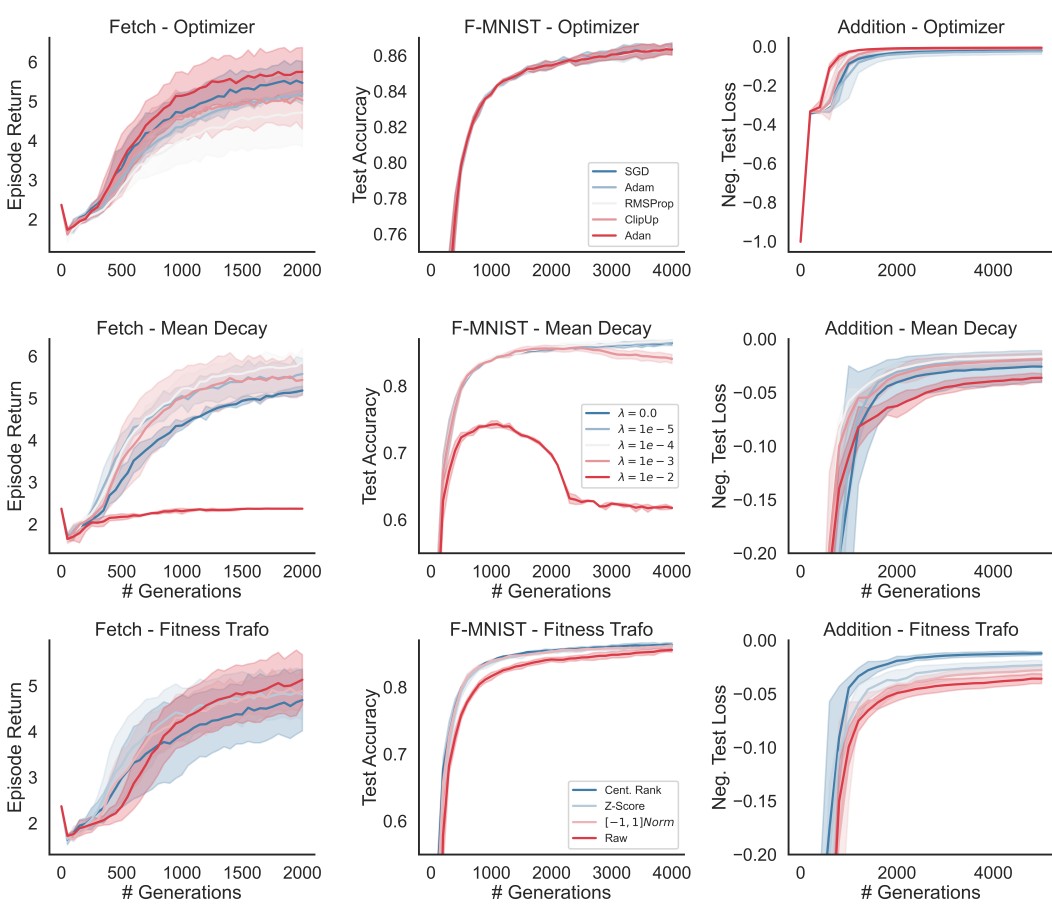

Figure 5: Impact of optimizer choice, mean decay and fitness transformation on OpenAI-ES performance. OpenAI-ES is largely robust to the optimizer choice, while decay and transformation are task-dependent. The results are averaged over 3 independent runs & we plot standard error bars.

one evaluate more parameter candidates at the cost of a potentially less informative performance estimate? Intuitively, this may be related to the inherent noise associated with the considered task. Noiseless tasks trivially do not require multiple evaluations, but how to assign evaluations in the face of uncertainty? We set out to investigate this question by introducing additive Gaussian noise to the return evaluation in the Ant control task. In Figure 4 we find that it is always beneficial to prefer a larger population over an increased number of evaluations – regardless of the external noise level. How do the gradient optimizer choice, mean multiplicative decay regularization and fitness normalization transformation affect the performance of OpenAI-ES? We consider a set of standard GD optimizers as well as ClipUp (Toklu et al., 2020), which was designed for EO. For Fetch Adan (Xie et al., 2022) outperforms the competitors by a slight margin. Otherwise, in Figure 5 we observe that OpenAI-ES performs robustly across optimizers for three of the NEB tasks.

In line with GD-based optimization we see that too much mean decay ($\theta_{t+1} = (1 - \lambda) \times \tilde{\theta}_{t+1}$) regularization can be hurtful, while small values generally improve test performance. Centered rank fitness transformation (Salimans et al., 2017) provides a good default setting, while for the Fetch task improvements can be achieved by using the raw return fitness score instead.

Finally, we investigate the scaling capabilities of OpenAI-ES for two key variables: Model and population size. In Figure 6 we evaluated the performance of control policies, CNN classifier and GRU networks for varying numbers of optimization parameters and increasing populations (given a fixed number of EO generations). We find that an increased population size always leads to better EO performance. An increase in model capacity, on the other hand, does not always lead to an improvement, which may be due to the inherent required task solution and the curse of dimensionality.

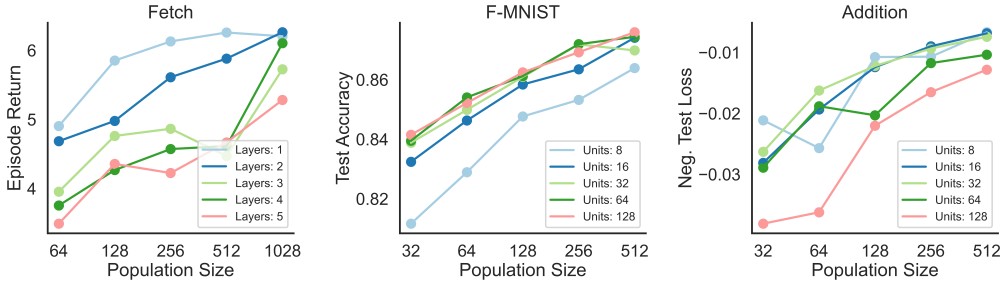

Figure 6: Scaling results for EO. We consider the performance behavior of OpenAI-ES with a varying number of population members and model size. Larger populations are always beneficial, while the required model capacity is task-dependent. The results are averaged over 3 independent runs.

## 5   Discussion

**Summary**. We introduce `NeuroEvoBench` – a benchmark targeting the rigorous evaluation of Evolutionary Optimization algorithms in the context of neural network training. It allows for easy, programmatic, and automatic investigation into a wide range of different BBO/neuroevolution tasks, types of EO, and budgets. Our initial investigation answers several core scientific questions regarding resource allocation, fitness-shaping, regularization, and scalability of EO.

**Limitations**. We aim to support a representative selection of tasks and problems. Nonetheless, we acknowledge that the actual real-world workload can represent a much more comprehensive range of scenarios. Going forward and as EO methods increase in capabilities, we envision a gradual enhancement and update of the task sweet and evaluation protocols.

**Ethical Considerations**. As a benchmark paper, we do not identify particular issues *per se*. Since we expect our work to be used by the community, we call for a range of considerations, including those of potential implicit bias in the task selection, evaluation protocols and result interpretation.

**Future Work**. We acknowledge that benchmarks have to be 'living projects' requiring constant refinement and updating with respect to the community's needs. We plan to continuously add more results for EO supported by `evosax` (Lange, 2022a) and to provide a modular and easy to extend benchmark protocol. Further investigations may include strategy restarts, asynchronous evaluation methods or the influence of shared randomness.

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
