# Appendix

## Table of Contents

# A  Full Task Descriptions

The following section reviews the proposed task sweep and optimization surrogates in more detail.

## A.1  BBO Tasks

We start by reviewing the standard black-box optimization tasks, which NEB incorporates for completeness. These include both the noiseless BBOB (Hansen, 2009) and HPO-B (continuous surrogate model) (Arango et al., 2021) tasks. BBOB consists of a set of functions with different properties, e.g. uni- versus multi-modal fitness landscapes with moderate or high conditioning and other structural properties (see Figure 7). The aim of the EO is to minimize the attained function value. HPO-B, on the other hand, considers the exercise of tuning hyperparameters of small Machine Learning models. The continuous version uses a gradient-boosted tree surrogate model in order to interpolate the performance from the originally discrete setting. For both settings, the EO methods have to optimize raw parameter vectors.

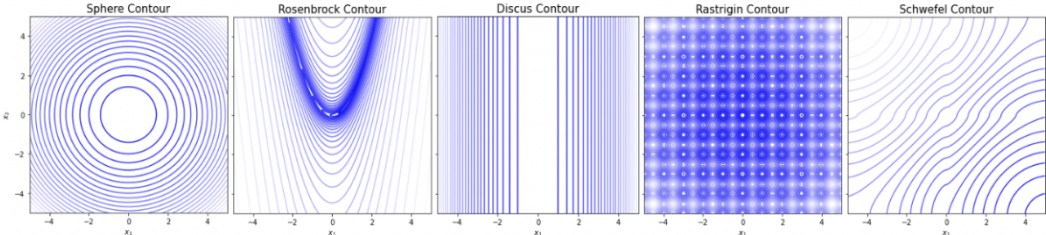

Figure 7: Example of 2-dimensional contour plots for BBOB (Hansen, 2009) functions.

| Parameter | Value |
|---|---|
| Population size | 32 |
| # Dimensions | <50 |
| # Generations | 100 |
| Fitness | Function value |
| Performance (max) | Aggregated neg. fct. value |
| EO init range | $[-5, 5]$ |
| # Evaluation runs | 50 |

Table 3: BBOB (Hansen, 2009).

| Parameter | Value |
|---|---|
| Population size | 4 |
| # Dimensions | <20 |
| # Generations | 100 |
| Fitness | Normalized task score |
| Performance (max) | Aggregated norm. task score |
| EO init range | $[0.5, 0.5]$ |

Table 4: HPO-B (Arango et al., 2021).

## A.2  Control Tasks

Next, we consider a set of control tasks, both robotic and visual pixel-based (see Figure 8). The agents are parametrized by deterministic policies (MLP with tanh output layer for robotic control and CNN with argmax output layer for visual control). The resulting parameter sets are evolved to maximize the cumulative episode return of the agents. We use the `brax` (Freeman et al., 2021) and `gymnax` (Lange, 2022b) libraries for fast accelerated policy rollouts. The `brax` tasks make use of observation normalization.

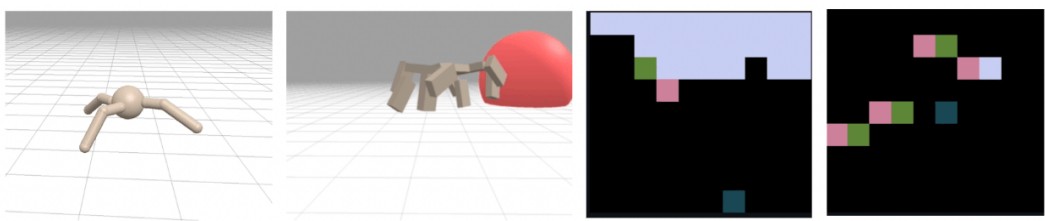

Figure 8: Control tasks including Brax (Freeman et al., 2021) and MinAtar (Young and Tian, 2019).

| Parameter | Value |
|---|---|
| Population size | 256 |
| # Generations | 2000 |
| Episode steps | 500 |
| MC evaluations/member | 8 |
| Hidden layers, units & activ. | 2, 32 & tanh |
| Fitness | Cum. ep. return |
| Performance (max) | Cum. ep. return |
| EO init range | $[0, 0]$ |

Table 5: Ant & Fetch (Freeman et al., 2021).

| Parameter | Value |
|---|---|
| Population size | 256 |
| # Generations | 1500 |
| Episode steps | 500 |
| MC evaluations/member | 8 |
| CNN layers, features & kernel | 1, 16 & (3, 3) |
| MLP layers, units & activ. | 1, 32 & ReLU |
| Fitness | Cum. ep. return |
| Performance (max) | Cum. ep. return |
| EO init range | $[0, 0]$ |

Table 6: Asterix & Breakout (Young and Tian, 2019; Lange, 2022b).

## A.3 Vision Tasks

The vision tasks consist of two classification problems (Fashion MNIST and CIFAR-10) as well as a VAE MNIST digit generation task. For the classification settings, we optimize the weights to minimize the training loss, i.e. cross-entropy loss, and evaluate the accuracy on the test set. For the VAE task we minimize the ELBO loss on the train dataset and evaluate the same loss on the test set.

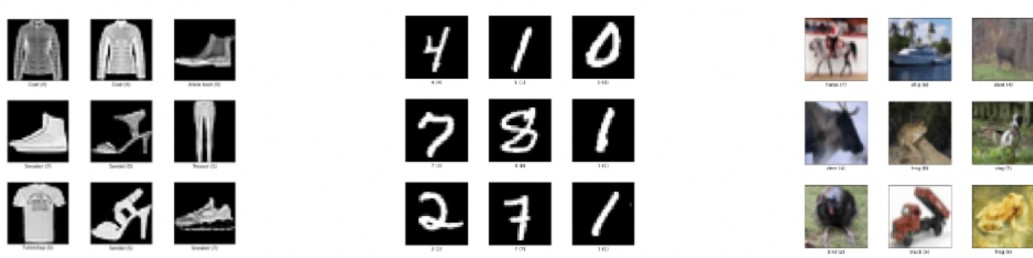

Figure 9: Vision tasks including F-MNIST/CIFAR-10 classification, MNIST generation.

| Parameter | Value |
|---|---|
| Population | 128 |
| # Generations | 400 |
| Batch size | 1024 |
| CNN | 2, 18/6 & (5, 5) |
| Fitness | Cross-entropy |
| Perf. (max) | Test accuracy |
| EO init range | $[0, 0]$ |

Table 7: F-MNIST Classification.

| Parameter | Value |
|---|---|
| Population | 256 |
| # Generations | 8000 |
| Batch size | 1024 |
| MLP Hidden U | 32 |
| VAE Latent U | 20 |
| Fitness | ELBO |
| Perf. (max) | Neg. ELBO |
| EO init range | $[0, 0]$ |

Table 8: MNIST Generation.

| Parameter | Value |
|---|---|
| Population | 128 |
| # Generations | 2500 |
| Batch size | 128 |
| Model Type | Small All-CNN |
| Fitness | Cross-entropy |
| Perf. (max) | Test accuracy |
| EO init range | $[0, 0]$ |

Table 9: CIFAR Classification.

## A.4 Sequence Tasks

Finally, we consider two sequence prediction tasks: The addition regression (Le et al., 2015) and the sequential MNIST classification task. For the addition setting, the recurrent network (GRU, Cho et al., 2014) receives a two-dimensional input consisting of a random floating point between 0 and 1 as well a binary indicator of whether to keep it in memory for an addition. Note that the binary indicator is only set to 1 for two timesteps. The final task is to predict the sum of two values. For sequential MNIST, on the other hand, the network (LSTM, Hochreiter and Schmidhuber, 1997) receives the individual pixels of MNIST digits sequentially. At the final timestep, it has to predict the digit identity. The sequence length is set to 150 and for the addition task and 784 for MNIST. Thereby, both tasks test the optimization of systems, which require long-term memory.

| Parameter | Value |
|---|---|
| Population | 128 |
| # Generations | 5000 |
| Batch size | 1024 |
| GRU units | 32 |
| Fitness | MSE loss |
| Perf. (max) | Neg. MAE loss |
| EO init range | $[0, 0]$ |

Table 10: Addition Regression.

| Parameter | Value |
|---|---|
| Population | 512 |
| # Generations | 3500 |
| Batch size | 512 |
| LSTM units | 48 |
| Fitness | Cross-entropy |
| Perf. (max) | Test accuracy |
| EO init range | $[0, 0]$ |

Table 11: S-MNIST Classification.

## B  Hyperparameter Tuning Ranges

- OpenAI-ES (Salimans et al., 2017) and PGPE (Sehnke et al., 2010)
  - Initial perturbation strength: $\sigma_0 \in [0.01, 0.15]$
  - Initial learning rate: $\alpha_0 \in [0.005, 0.05]$
  - Gradient descent optimizer: Adam (Kingma and Ba, 2014)
  - Fitness transformation: centered ranks
  - Exponential learning rate decay: 0.999
  - Exponential perturbation strength decay: 0.999

- ARS (Mania et al., 2018): Same as OpenAI-ES/PGPE, but we also tune the fitness transform.
  - Fitness transformation $\in$ {raw, z-score, ranks}

- SNES (Schaul et al., 2011): Next to the default fitness shaping, SNES can also use the following utility/fitness shaping transformation as proposed by Lange et al. (2022):

$$\boldsymbol{w}_{t,j} = \text{softmax}\left(\beta \times (\text{rank}(j)/N - 0.5)\right), \ \forall j = 1, ..., N.$$

  - Initial perturbation strength: $\sigma_0 \in [0.01, 0.15]$
  - Softmax sharpness/temperature: $\beta \in [10, 15, 20, 25, 30, 35, 40]$

- Sep-CMA-ES (Ros and Hansen, 2008)
  - Initial perturbation strength: $\sigma_0 \in [0.01, 0.15]$
  - Elite ratio $\in$ {0.1, 0.2, 0.3, 0.4, 0.5}

- LES (Lange et al., 2022)
  - Initial perturbation strength: $\sigma_0 \in [0.01, 0.15]$

- Simple GA (Rechenberg, 1978)
  - Initial perturbation strength: $\sigma_0 \in [0.01, 0.15]$
  - Elite ratio $\in$ {0.0, 0.1, 0.2, 0.3, 0.4, 0.5}

- SAMR-GA (Clune et al., 2008)
  - Initial perturbation strength: $\sigma_0 \in [0.01, 0.15]$
  - Elite ratio $\in$ {0.0, 0.1, 0.2, 0.3, 0.4, 0.5}
  - Meta-perturbation strength for $\sigma$: $\in [1.0, 3.0]$

- GESMR-GA (Kumar et al., 2022)
  - Initial perturbation strength: $\sigma_0 \in [0.01, 0.15]$
  - Elite ratio $\in$ {0.0, 0.1, 0.2, 0.3, 0.4, 0.5}
  - Meta-perturbation strength for $\sigma$: $\in [1.0, 3.0]$

- LGA (Lange et al., 2023)
  - Initial perturbation strength: $\sigma_0 \in [0.01, 0.15]$
  - Elite ratio $\in$ {0.0, 0.1, 0.2, 0.3, 0.4, 0.5}

## C  Fitness Shaping Transformations

- Centered ranks transformation: $\tilde{f}_i = \texttt{rank}(f_i|\{f_j\}_{j=1}^N)/N - 0.5$ with $\texttt{rank}(f_i|\{f_j\}_{j=1}^N) \in \{0, 1, \ldots, N-1\}$. The best-performing population member has rank 0.

- Z-score: $\tilde{f}_i = (f_i - \mu(\{f_j\}_{j=1}^N))/\sigma(\{f_j\}_{j=1}^N)$ with fitness mean $\mu$ & standard deviation $\sigma$

- $[-1, 1]$ range normalization: $\tilde{f}_i = 2 \times \frac{f_i - \min\{f_j\}_{j=1}^N}{\max\{f_j\}_{j=1}^N - \min\{f_j\}_{j=1}^N} + \min\{f_j\}_{j=1}^N$

## D  Code, Data Availability & Compute Resources

**Code & Documentation**. The `NeuroEvoBench` codebase is open-sourced under Apache 2.0 license and publicly available under `https://github.com/neuroevobench/neuroevobench`. The accompanying analysis and experiment configuration can be found under `https://github.com/neuroevobench/neuroevobench-analysis`. Furthermore, we provide a documentation webpage for further information and result summarization: `https://sites.google.com/view/neuroevobench`. Finally, we provide a colab notebook, which explains how to add a custom EO method and executes a random search pipeline for the BBOB tasks. It can be found here: `https://colab.research.google.com/github/neuroevobench/neuroevobench/blob/main/examples/neb_introduction.ipynb`.

All training loops and ES are implemented in JAX (Bradbury et al., 2018). All visualizations were done using Matplotlib (Hunter, 2007) and Seaborn (Waskom, 2021, BSD-3-Clause License). Finally, the numerical analysis was supported by NumPy (Harris et al., 2020, BSD-3-Clause License). Furthermore, we used the following libraries: Evosax: Lange (2022a), Gymnax: Lange (2022b), Evojax: Tang et al. (2022), Brax: Freeman et al. (2021).

**Data Availability**. We make all data used in our experiments publicly available in a Google Cloud Storage bucket. This includes all random search sweeps for 10 EO methods on the 9 considered tasks as well as the multi-seed re-evaluations. The data can be downloaded by executing `gsutil -m -q cp -r gs://neuroevobench/ .`. All figures displayed in this paper can then be reproduced by executing the accompanying notebooks provided here: `https://github.com/neuroevobench/neuroevobench-analysis`. We hope that this will enable other researchers to directly benchmark their methods against the 10 baselines without having to spend substantial compute on re-evaluating the other methods.

**Compute Requirements & Experiment Organization**. The experiments were organized using the `MLE-Infrastructure` (Lange, 2021, MIT license) training management system.

Simulations were conducted on a high-performance cluster using between 1 and 5 independent runs (random seeds). We mainly rely on individual V100S and A100 NVIDIA GPUs.

The individual random search experiments last between 4.5 and 50 hours depending on the considered task and EO combination. The final evaluation of a tuned configuration, on the other hand, only requires up to a single hour.

The tasks were chosen so that executing the entire benchmark only requires 2.5 days given 9 suitable GPUs. Furthermore, the open-data availability of the benchmark results allows researchers to focus on their method, instead of having to spend computing resources in order to collect baseline results.

# E  Additional Results

## E.1  Aggregated Performance by Problem Class

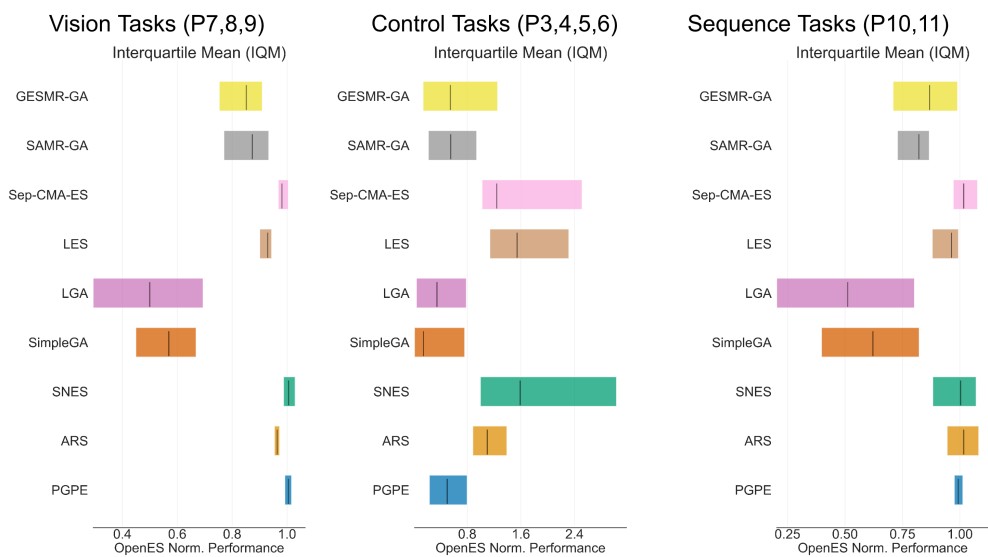

Figure 10: Aggregated normalized median performance across EO algorithms grouped by the problem class (vision, control, sequence).

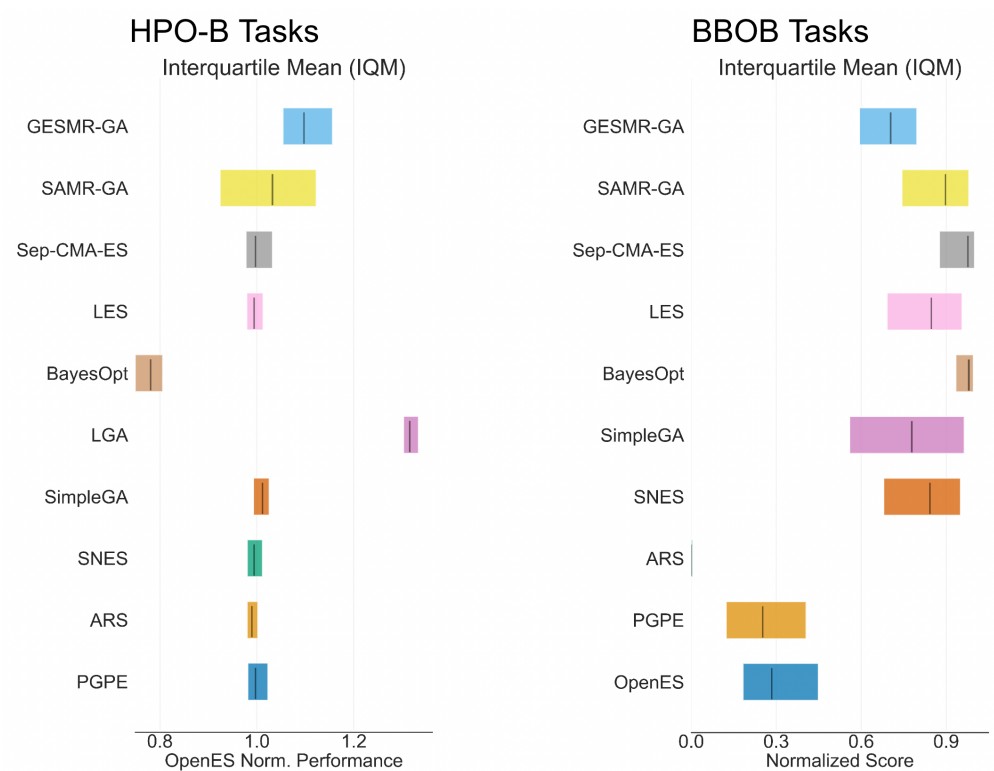

Figure 11: Aggregated normalized median performance across EO algorithms on BBOB and HPO-B tasks.

## E.2 Hyperparameter Robustness of EO Methods across Tasks

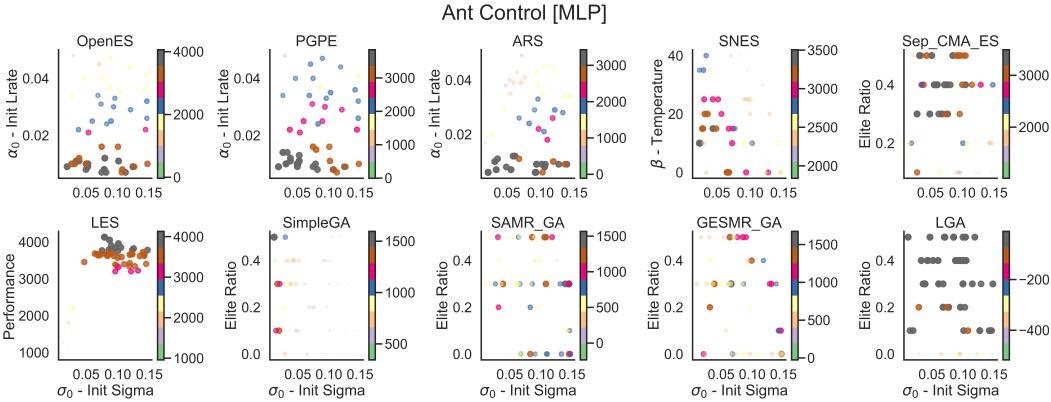

Figure 12: Ant (P3) control task hyperparameter robustness across EO methods.

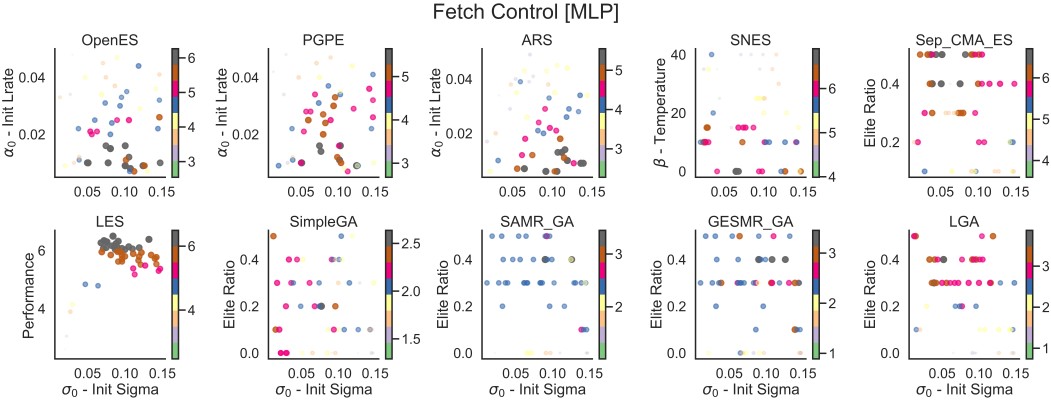

Figure 13: Fetch (P4) control task hyperparameter robustness across EO methods.

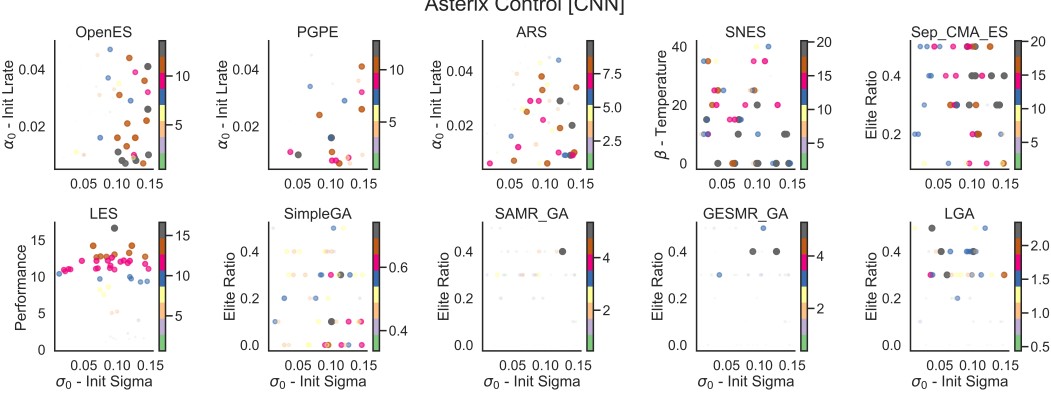

Figure 14: Asterix (P5) control task hyperparameter robustness across EO methods.

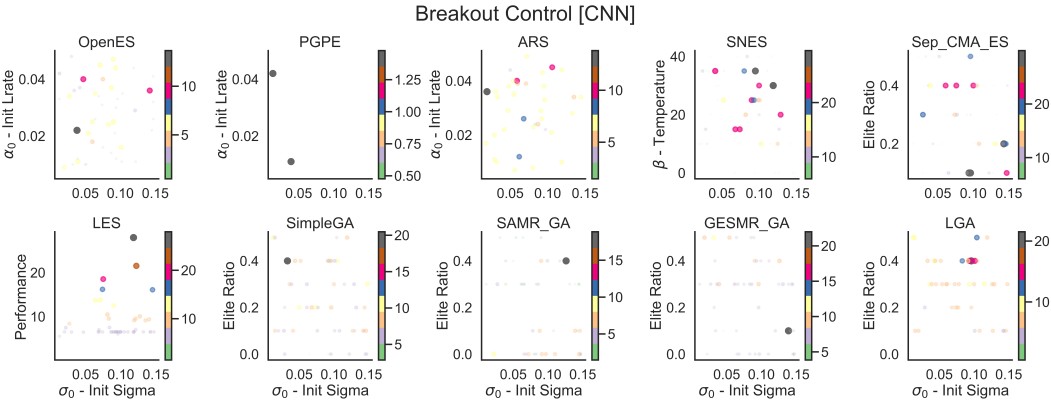

Figure 15: Breakout (P6) control task hyperparameter robustness across EO methods.

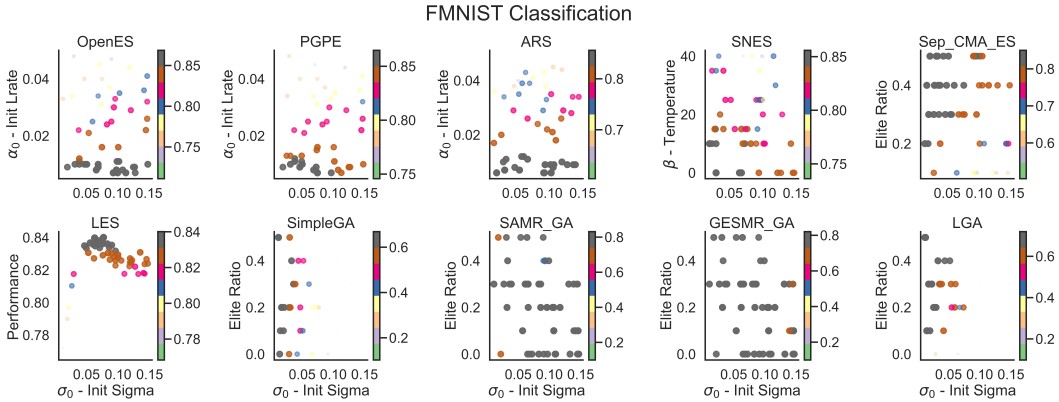

Figure 16: F-MNIST Classification (P7) vision task hyperparameter robustness across EO methods.

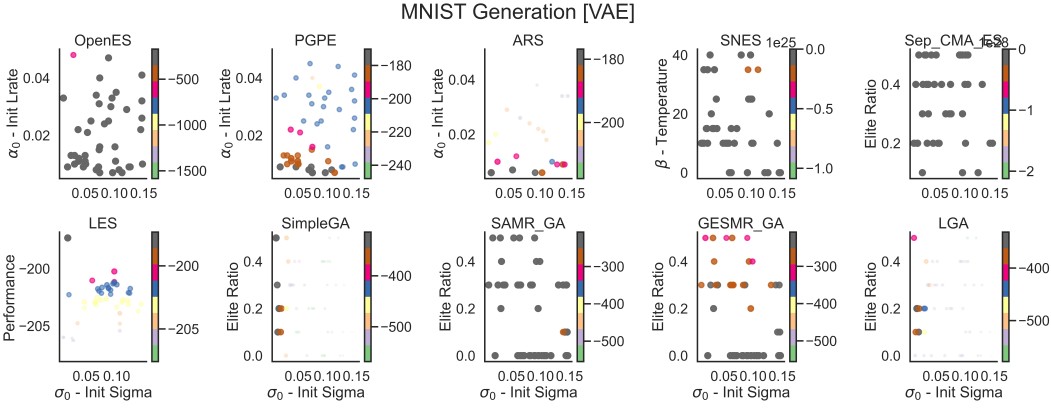

Figure 17: MNIST Generation (P8) vision task hyperparameter robustness across EO methods.

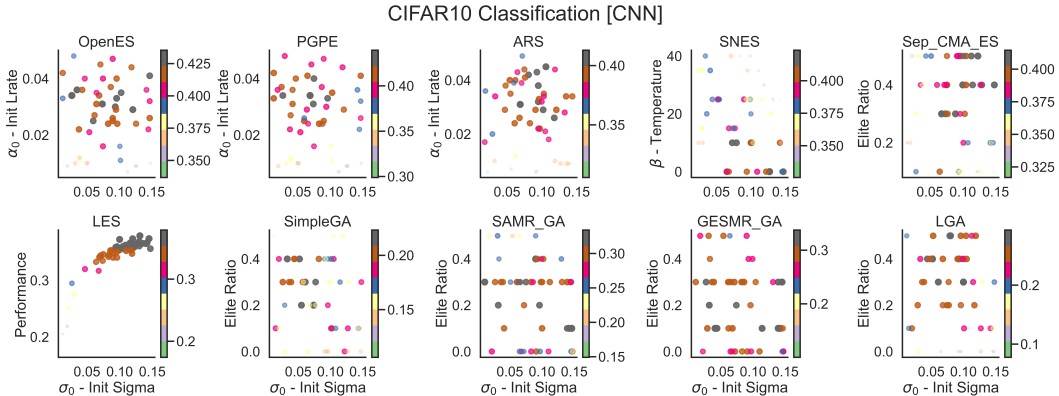

Figure 18: CIFAR-10 Classification (P9) vision task hyperparameter robustness across EO methods.

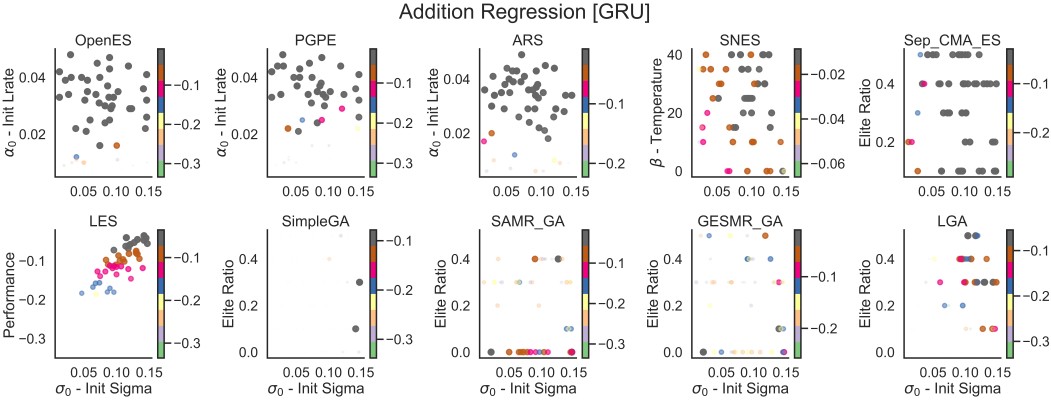

Figure 19: Addition (P10) sequence task hyperparameter robustness across EO methods.

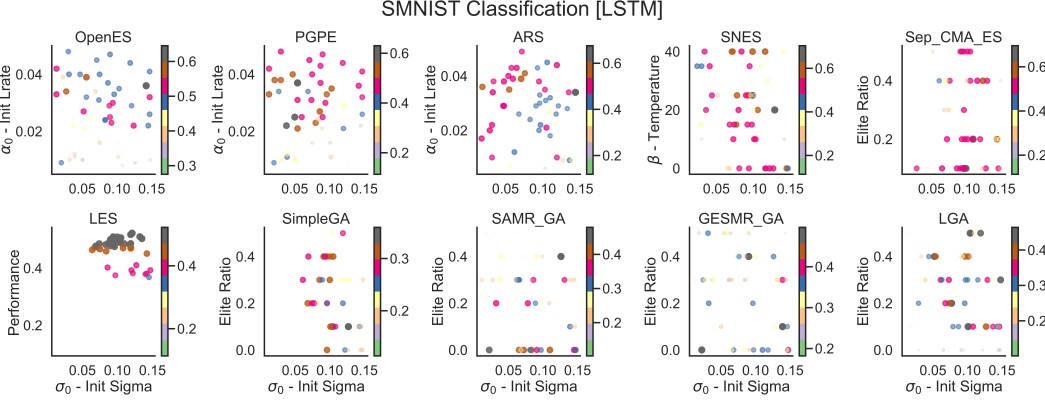

Figure 20: S-MNIST classification (P11) sequence task hyperparameter robustness across EO methods.