# OpenReview forum: "NeuroEvoBench:  Benchmarking Evolutionary Optimizers for Deep Learning Applications"
_NeurIPS.cc/2023/Track/Datasets_and_Benchmarks — NeurIPS 2023 Datasets and Benchmarks Poster_

### Official Review · Reviewer_geZa · 2023-07-04
**The authors introduced a benchmark to evaluate Evolutionary Optimization algorithms for neural network training.**

**Rating:** 7
**Confidence:** 1
**Clarity:** Yes

**Strengths:**

--The work describes very well the important challenge of benchmarking evolutionary optimizations with real-world data and state-of-the-art optimization methods.
--The paper is very well written and structured.
--The introduction of the motivation is thorough.
--The curation of the different datasets has been thoughtful and complete.

**Additional Feedback:**

No

**Correctness:**

I got the error when I tried to run the colab notebook by the author: "AttributeError: module 'PIL' has no attribute 'Image'"

**Documentation:**

The size of each dataset was not mentioned.

**Limitations:**

Yes

**Opportunities For Improvement:**

The benchmark against Bayesian optimization is not included or discussed.

**Relation To Prior Work:**

Yes

**Summary And Contributions:**

This benchmark presented the tasks, experiment protocols, and evolutionary optimizer wrappers for evaluating the performance of new gradient-free optimization methods such as Evolution Strategies and Genetic Algorithms. This benchmark focuses on non-synthetic datasets to facilitate benchmarking of evolutionary optimization methods. All task evaluations are written in JAX to save evaluation time.

---

> ### Author Response · Authors · 2023-08-17
> **Reply - Reviewer geZa**
>
> We thank the review for their response and acknowledgement of our additional work, and especially for recognizing that our paper "is very well written and structured." and its "curation of the different datasets has been thoughtful and complete."
>
> > “The benchmark against Bayesian optimization is not included or discussed.”
>
> Thanks for the suggestion of benchmark regarding comparison with BO (Bayesian optimization) and we agree this could be a helpful addition. We have added a Bayesian optimization baseline to the benchmark repository (see [Github repo](https://github.com/neuroevobench/neuroevobench/blob/main/neuroevobench/blines/bayes_opt_jax.py)) and will incorporate the experiment results shortly to the manuscript.
>
> In a wider context, BO is commonly applied to hyperparameter optimization problems with few parameters to be optimized and single sequential evaluations. While there are workarounds including inducing points, and batch versions etc. but generally speaking, BO is not an out-of-the-box applicable optimization method of neural network tasks with relatively large amounts of parameters. We will add this discussion in the revision of our manuscript.
>
> > “I got the error when I tried to run the colab notebook by the author: "AttributeError: module 'PIL' has no attribute 'Image'"
>
> Thanks for trying out our notebook and we excuse the inconvenience. After examination, we found that there seems to have been a dependency error and we have addressed the issue. The notebook should run without any problems.

---

> > ### Comment · Reviewer_geZa · 2023-08-29
> >
> > The authors have addressed my concerns, therefore I would like to keep my score.

---

### Official Review · Reviewer_1pB9 · 2023-07-19
**a benchmark of evolutionary optimizers**

**Rating:** 7
**Confidence:** 3
**Correctness:** yes
**Clarity:** see above

**Strengths:**

the experiments are broad. the authors repot on an interesting topic. copmaring across many tasks is nice, as well as the sensitivity analysis.

**Additional Feedback:**

- neuroevolution traditionally refers to evolutionary computation approaches that adapt the architecture of a network, not just its parameters. the title and references to neuroevolution do not make clear that the authors are addressing the narrower problem of parameter optimization.

- Machine Learning does not need to be written in capital letters.

- the authors sometimes say machine learning when they specifically mean deep learning. please use the appropriate (narrower) language.

- on GAs "they keep an archive of 'parent' solutions from which new 'children' candidates are sampled and adapted with mutation."
 - are any of these GAs doing crossover?

**Documentation:**

yes

**Limitations:**

yes

**Opportunities For Improvement:**

the following are aspects of the paper that I think need improvement:

- presentation of results
  - the figures 2,3, and 4 are hard to read and not color-blind friendly. The results could be summarized much more clearly, for example by showing final model performance comparisons and calculating ranking statistics. Visual comparisons are generally difficult to interpret.

- discussion of prior work
  - the paper doesn't discuss any prior work to benchmark evolutionary optimizers for neural networks.
  - the authors should differentiate their contribution from, e.g., the benchmark work done with nevergrad

**Relation To Prior Work:**

see above

**Summary And Contributions:**

the authors present a benchmark of some evolutionary optimization methods on black box optimization and neural network weight optimization. they compare 10 EOs on 4 benchmark sets: BBO, control, vision, and sequence prediction. they evaluate sensitivity to model size, population size and noise.

---

> ### Author Response · Authors · 2023-08-17
> **Reply - Reviewer 1pB919**
>
> We thank the reviewer for their comments and appreciation. We would like to take the opportunity to discuss their suggestions.
>
> > “[...] the figures 2,3, and 4 are hard to read and not color-blind friendly. The results could be summarized much more clearly, for example by showing final model performance comparisons and calculating ranking statistics. Visual comparisons are generally difficult to interpret.”
>
> We thank the reviewer for raising awareness of the color palette. In our initial submission we have already aimed to address such concern using a colorblind-suited color palette (see [colab notebook](https://github.com/neuroevobench/neuroevobench-analysis/blob/main/analysis/02_lcurves.ipynb)), but please let us know if you recommend a specific palette.
> Furthermore, regarding ranking statistics, In our initial submission we already provide an aggregation of all task-specific results in figure 1 (right plot) using the interquartile mean calculated on the final performance of the different optimizers.
>
> > “[...] discussion of prior work: the paper doesn't discuss any prior work to benchmark evolutionary optimizers for neural networks. [...] the authors should differentiate their contribution from, e.g., the benchmark work done with nevergrad.”
> To the best of our knowledge there does not seem to exist a directly comparable benchmark. That said, there exist a few loosely related works worthy mentioning:
>
> One being nevergrad [1], as the reviewer pointed out. The other being Vizier [2]. However, the associated benchmarks are solely considering BBO hyperparameter tuning and the common BBOB functions. Hence, they do not consider very high-dimensional neural network tasks.
>
> Furthermore, Mousavirad et al. [1] concerns gradient-free optimization, but it
> - only investigates population-based metaheuristic algorithms and does not include the popular set of considered evolution strategies which is the focus of our study.
> - Unfortunately, it is closed-source, not easily accessible by the community as well as us.
>
> We added a discussion of nevergrad, vizier and Mousavirad et al. in the related work section for  better clarification.
>
> > “neuroevolution traditionally refers to evolutionary computation approaches that adapt the architecture of a network, not just its parameters. the title and references to neuroevolution do not make clear that the authors are addressing the narrower problem of parameter optimization.”
>
> We thank you for the suggestion on improving the title. Would the following title address your concerns? “NeuroEvoBench: Benchmarking Evolutionary Optimizers for Deep Learning Applications”
>
> > “Machine Learning does not need to be written in capital letters. [...] the authors sometimes say machine learning when they specifically mean deep learning. please use the appropriate (narrower) language.”
>
> We have revised the manuscript accordingly, fixing the term as you kindly pointed out. More specifically, we adopt the Deep Learning term. Thanks.
>
> > “on GAs "they keep an archive of 'parent' solutions from which new 'children' candidates are sampled and adapted with mutation."
>
> We have revised the manuscript by correcting the sentence.Thanks.
>
> > “are any of these GAs doing crossover?”
>
> No, they do not, for the following reasons:
> - Preliminary experiments indicated that crossover in neural network weight space is fairly ineffective and did not add any performance gains.
> - Furthermore, SAMR-GA and GESMR-GA do not implement cross-over.
>
> In our revision we added a sentence clarifying this detail.
>
> We hope that our reply could address some of your concerns. Please let us know if any further questions may have come up.
>
> References:
>
> [1] Mousavirad, Seyed Jalaleddin, et al. "A benchmark of recent population-based metaheuristic algorithms for multi-layer neural network training." Proceedings of the 2020 genetic and evolutionary computation conference companion. 2020.
>
> [2] Golovin, D., Solnik, B., Moitra, S., Kochanski, G., Karro, J., & Sculley, D. (2017, August). Google vizier: A service for black-box optimization. In Proceedings of the 23rd ACM SIGKDD international conference on knowledge discovery and data mining (pp. 1487-1495).
>
> [3] Bennet, P., Doerr, C., Moreau, A., Rapin, J., Teytaud, F., & Teytaud, O. (2021). Nevergrad: black-box optimization platform. ACM SIGEVOlution, 14(1), 8-15.

---

> > ### Comment · Reviewer_1pB9 · 2023-08-29
> >
> > > We thank the reviewer for raising awareness of the color palette. In our initial submission we have already aimed to address such concern using a colorblind-suited color palette (see colab notebook), but please let us know if you recommend a specific palette.
> >
> > I can tell you anecdotally as a color-blind person (n of 1) that I cannot tell all of these colors apart. I recommend using line style or marker styles to differentiate the plots based on something other than just color.
> >
> > > Furthermore, regarding ranking statistics, In our initial submission we already provide an aggregation of all task-specific results in figure 1 (right plot) using the interquartile mean calculated on the final performance of the different optimizers.
> >
> > I see it now, thanks. It would be good to move this into the results section rather than pairing it with the diagram, if space allows.
> >
> > > We added a discussion of nevergrad, vizier and Mousavirad et al. in the related work section for better clarification.
> >
> > That clarifies things, thanks.
> >
> > > We thank you for the suggestion on improving the title. Would the following title address your concerns? “NeuroEvoBench: Benchmarking Evolutionary Optimizers for Deep Learning Applications”
> >
> > Yes, this is much more clear.
> >
> > Based on the author's response I plan to update my score.

---

### Official Review · Reviewer_ezSc · 2023-07-21
**NeuroEvoBench: Benchmarking Neuroevolution for Large-Scale Machine Learning Applications**

**Rating:** 7
**Confidence:** 2

**Strengths:**

Numerous comprehensive experiments were conducted involving 11 different problems, various optimizers, and diverse fitness and experiment types. Despite being in its early stages, this research holds great potential for benefiting the scientific community in the future.

**Additional Feedback:**

Please revisit the control section again.

**Clarity:**

Yes the paper is well wriiten and all technical aspects clearly in the manuscript and supplementary material.

**Correctness:**

The paper looks technically correct and experiments are performed to back the claims

**Documentation:**

Yes

**Ethics:**

It is a combination of existing datasets and benchmarks and no visible ethical violations is noted.

**Limitations:**

Though enough experimentations were done but main concern is lack of insight in the data and detail investigation is needed
specifically for the control group. We see deviant behavior in the fetch control group in figures 2 and 5 and would like further insight
Referring fig 4 did more experiments conducted on higher noise levels?


**Opportunities For Improvement:**

Since it is early stage so more experiments, problems and combinations need to be added for a more comprehensive benchmark.

**Relation To Prior Work:**

Since this is one of the early works connecting EO with deep learning experiments, the existing research gap is explained nicely and need for a new dataset is established.

**Summary And Contributions:**

The primary objective of this study is to establish a novel benchmark for Evolutionary Optimization (EO) techniques, specifically designed for deep learning applications. The research extensively assesses both conventional and meta-learned evolutionary optimizers. The main contribution of this work lies in the development of a hardware-accelerated EO benchmark, which fills a critical need in the field, as hardware-based accelerators have become increasingly popular. This benchmark dataset is essential for the scientific community as it enables standardized and comprehensive benchmarking of EO methods in the context of deep learning.

---

> ### Author Response · Authors · 2023-08-17
> **Reply - Reviewer ezSc**
>
> We thank the reviewer for their suggestions, appreciation of the manuscript writing and improvement recommendations, and especially for recognizing that our paper “this research holds great potential for benefiting the scientific community in the future.” We address the reviewer’s concerns in the following.
>
> > “Since it is early stage so more experiments, problems and combinations need to be added for a more comprehensive benchmark.”
> In our revision, we have now added a set of additional experiment results on the BBOB and HPO-B tasks in the supplementary material (see Figure 11) and further added a Bayesian Optimization baseline to the benchmark repository.
>
> While more problems can potentially provide more insights, we also have to consider a parsimonious design which facilitates usage by the broader community. This includes strategically limiting the amount of required compute resources for investigating a new method by a user in the community. Therefore, we curate a careful selection of representative tasks to balance the potential insight and resource requirements.
>
> > “Though enough experimentations were done but main concern is lack of insight in the data and detail investigation is needed specifically for the control group. We see deviant behavior in the fetch control group in figures 2 and 5 [...]”
> In our revision, we have now added more details regarding the top performing settings from the random search experiments for all tasks and optimizers in the supplementary material.
>
> Deviant behaviors reported in figure 2 and 5 arise from different hyperparameter settings for OpenAI-ES that are purposely chosen:
>  In figure 2 we fix the optimizer to Adam, mean/weight decay to 0 and use the centered rank transformation. These are the common default settings for such parameters and we only perform random search over the initial perturbation strength and learning rate.
> In figure 5, on the other hand, we wanted to further explore the optimizer choice, mean decay and fitness transformation.
>
> > “[...] and would like further insight Referring fig 4 did more experiments conducted on higher noise levels?”
>
> We intend Figure 4 to answer the question of resource allocation as it shows an interesting phenomenon that regardless of the amount of noise, an increase in the population size is more effective than an equal increase in Monte Carlo evaluations. This can be best seen along the diagonal elements of the heat plots and was consistent across noise levels.
>
> We didn’t consider extreme noise values yet, but expect the performance to drop once the fitness signal is drowned by the noise.

---

> > ### Comment · Reviewer_ezSc · 2023-08-29
> >
> > Thank you for the explanation

---

### Official Review · Reviewer_YBAt · 2023-07-24
**The paper lays a solid foundation with its benchmarking study and insights into EO methods but the target is not clear.**

**Rating:** 5
**Confidence:** 3
**Clarity:** The paper is well written and easy to…

**Strengths:**

The paper provide a benchmarking study of evolutionary optimization (EO) methods for large-scale machine learning applications, covering a wide range of tasks, types of EO, and budgets.

The papers provide insights into the challenges and opportunities of using EO methods.

The benchmarking methodology is rigorous and well-documented.

**Additional Feedback:**

None

**Correctness:**

Regarding the motivation for lack of parallelization, it's worth noting that the high-performance computing community has made significant progress in addressing this issue. Multiple open-source libraries, such as Optuna and Ray, already offer parallelization capabilities for EO methods. However, an intriguing direction for further exploration could be to focus on the asynchronous aspect of EO and examine the influence of randomness and varying evaluation times within the population.


**Documentation:**

The documentation and the repo are presented with sufficient detail. There is sufficient detail to support reproducibility.

**Ethics:**

The authors discuss potential negative societal impacts of their work and call for a range of considerations, including those of potential implicit bias in the task selection, evaluation protocols, and result interpretation.


**Limitations:**

Please see the feedback on Opportunities For Improvement. The potential negative societal impact of the work is rather minor.

**Opportunities For Improvement:**

The target ML applications for this study could be clarified further. If the focus is on hyperparameter tuning, it's important to consider existing benchmarking papers that include the proposed set of algorithms. On the other hand, if the study revolves around specific benchmarks, it would be beneficial to explicitly mention and justify their choice.

To strengthen the paper, more clarity is needed on the characteristics, representativeness, and generality of the 11 presented problems. Understanding why these problems are relevant and what aspects make them suitable for EO algorithms would be insightful. Additionally, it is essential to justify the inclusion of non-EO algorithms in the study to ensure a broader perspective. The study's applicability and impact can be better evaluated by comparing EO methods with other gradient-based or model-based approaches such as Bayesian optimization. For example, what happens if one uses these benchmarks to show that their EO is the best but that is not competitive to other gradient based methods or model-based methods such as Bayesian optimization.

An essential aspect that requires careful consideration in the presented study is the characterization of the search space difficulty. It would be valuable for the paper to provide insights on landscape complexity to assess how challenging it is to navigate for different optimization methods. The authors can better justify why these benchmarks are well-suited targets for benchmarking EO algorithms, ultimately adding depth and significance to the paper.



**Relation To Prior Work:**

The related work is good.

**Summary And Contributions:**

The paper provides i) comprehensive benchmarking study exploring evolutionary optimization (EO) for large-scale machine learning applications; ii) valuable insights into challenges and opportunities of using EO methods; iii) detailed description of benchmarking methodology; and iv) thorough analysis of results from experiments on diverse datasets and models.

---

> ### Author Response · Authors · 2023-08-17
> **Reply - Reviewer YBAt (Part 1)**
>
> We thank the reviewer for their time, substantial feedback and insightful questions. We also thank the reviewer for recognizing that our paper “provides insights into the challenges and the opportunities of using EO methods” and that our “benchmarking methodology is rigorous and well-documented”. We updated the draft (the revisions in the paper are marked in red) and address the reviewer’s concerns in the following:
>
> >“The target ML applications for this study could be clarified further. If the focus is on hyperparameter tuning, it's important to consider existing benchmarking papers that include the proposed set of algorithms. On the other hand, if the study revolves around specific benchmarks, it would be beneficial to explicitly mention and justify their choice.”
>
> We agree that the target audience and applications can be better specified. Indeed, we focus on tasks where evolutionary methods are used to optimize neural network parameters and only added the BBOB and HPO-B task sweep for completeness.
>
> We provide clarification in the introduction and main text: *“[...] a plethora of challenging high-dimensional optimization problems where GD methods are inadequate exist, including not only hyperparameter search but also the optimization of non-differentiable operators (e.g. objective or architecture), the computation of ill-behaved gradients through long computational graphs, and applications requiring black-box optimization.”* (page 1).
>
> > “To strengthen the paper, more clarity is needed on the characteristics, representativeness, and generality of the 11 presented problems. Understanding why these problems are relevant and what aspects make them suitable for EO algorithms would be insightful.”
>
> We have added supplementary figure 12, which also evaluates the considered methods on the smaller BBOB and HPO-B tasks. The results show that the small benchmarks are insufficient in predicting the performance on neural network tasks. Hence, they provide further motivation for the tailored benchmark introduced by us. The remaining 9 tasks were chosen to cover a wide range of settings of potential interest. At the current point in time, evolutionary methods are most applicable to tasks which either require surrogate objectives to be solved with gradient descent (e.g. the selected RL tasks which don’t directly optimize the agent’s return [4 control tasks]) or settings which would have to propagate gradients through long unrolled computation graphs (e.g. sequence modeling with RNNs or meta-learning [2 sequence tasks]). Finally, as the software and hardware stack develops, settings with more parameters (e.g. vision models)  will become more and more feasible for EO methods to be applicable [3 vision tasks]. These considerations all influenced the choice of tasks.
>
> Again, we further added clarification in the main text: *“[...] Our task selection is motivated by the observation that small-scale BBO benchmarks alone (e.g. BBOB/HPO-B) do not suffice in predicting the performance of EO methods on high-dimensional tasks requiring the optimization of network weights (see comparison of Figures 11 and12). Furthermore, the different task classes cover a wide range of representative Deep Learning problems required for robust performance evaluation (see Figure 11).”*  (page 5).
>
> >“The study's applicability and impact can be better evaluated by comparing EO methods with other gradient-based or model-based approaches such as Bayesian optimization.”
>
> Thank you for raising this point. For now we chose to focus on algorithms inspired by evolutionary computation. Given that we consider tasks with >50k parameters, GP/covariance matrix estimation tends to become very hard, both statistically and computationally. We, therefore, already chose to neglect full-diagonal covariance ES. Bayesian Optimization (BO) is classically applied to hyperparameter optimization problems with small-medium dimensionality and single sequential evaluations. Of course there are workarounds including inducing points, and batch versions etc. but BO is arguably not out of the box applicable for the optimization of neural network tasks. We still chose to add a simple batch BO baseline to the benchmark repository (see [Link](https://github.com/neuroevobench/neuroevobench/blob/main/neuroevobench/blines/bayes_opt_jax.py)) and are waiting for the results of the experiments. We will report back once the results are completed.

---

> > ### Author Response · Authors · 2023-08-17
> > **Reply - Reviewer YBAt (Part 2)**
> >
> > >“An essential aspect that requires careful consideration in the presented study is the characterization of the search space difficulty. It would be valuable for the paper to provide insights on landscape complexity to assess how challenging it is to navigate for different optimization methods.”
> >
> > We fully agree. For our main experiments we followed the general benchmark protocol proposed by Schmidt et al. (2020), which uses a random search with space boundary refinement. This makes it hard to create nice visualizations for the robustness of the hyperparameters. Instead, we opted to perform a couple target grid searches in the experiment section 4.3. We have now also added scatter plots visualizing the hyperparameter robustness for each problem and evolutionary optimizer combination to the supplementary material (see Figures 13-21). We find that while the optimal set of hyperparameters varies across problems and EO, finite-difference-based ES methods largely share good parameter ranges. Furthermore, all results can be inspected in the openly available benchmark dataset.
> >
> > >“Regarding the motivation for lack of parallelization, it's worth noting that the high-performance computing community has made significant progress in addressing this issue. Multiple open-source libraries, such as Optuna and Ray, already offer parallelization capabilities for EO methods.”
> >
> > Yes, you are correct and we have added a note in the paper manuscript. Nonetheless, in our experience Optuna, Ray and Dask all require engineering overhead, e.g. setting up worker and orchestrator nodes, define how they collaborate, etc. since every problem differs. Furthermore, Ray can have odd hardware and cluster-specific idiosyncrasies. Our proposed benchmark task implementations leverage JAX, which comes with a set of minimal function transformations used to either auto-vectorize or device-parallelize fitness evaluations across the population. These are largely agnostic and robust, which circumvents a lot of the engineering effort.
> >
> > >“However, an intriguing direction for further exploration could be to focus on the asynchronous aspect of EO and examine the influence of randomness and varying evaluation times within the population.”
> >
> > We thank the reviewer for pointing out these exciting directions for further investigation which we fully agree with. While in this study we focus on the standard synchronous population evaluation setting and aim to establish a general benchmark allowing for flexible extension and comparative experimentation, we fully agree that it opens doors to many more questions including optimizer restarts and the usage of subpopulations.
> >
> > We revised the future work section accordingly to discuss them: *“[...] We plan to continuously add more results for EO supported by evosax and to provide a modular and easy to extend benchmark protocol. Further investigations may include strategy restarts, asynchronous evaluation methods or the influence of shared randomness.”* (page 9).
> >
> > We hope that our reply could address some of your concerns. Please let us know if any further questions may have come up. The authors.

---

### Author Response · Authors · 2023-08-17
**Rebuttal summary & general comments to all reviewers**

We thank the reviewers for their insightful comments and suggestions. We are happy to learn that the reviewers value our contribution and think that our work “holds great potential for benefiting the scientific community in the future” and that “the curation of the different datasets has been thoughtful and complete”.

We have responded to each reviewer separately. All additional results are summarized in the following [set of slides](https://docs.google.com/presentation/d/1okxaIes937RO7017bLYUjqhmbakIbqRImwqrTQZUAOs/edit?usp=sharing). Furthermore, we summarize our responses and additional work as follows:

For Reviewer YBAt24 ([link](https://openreview.net/forum?id=s6qtLyR6uJ&noteId=mGEEMvSVLQ)), we

1. Answered all the reviewer’s questions.
2. Added a Batch Bayesian Optimization baseline to the benchmark (see [GitHub link](https://github.com/neuroevobench/neuroevobench/blob/main/neuroevobench/blines/bayes_opt_jax.py)).
3. Added a set of visualizations to illuminate the hyperparameter robustness of the evolutionary optimization methods used in the benchmark study (see new figures 13-21).
4. Added a set of experiment results on non-neural network tasks to highlight the need for a separate benchmark tailored to the Deep Learning community and clarified the selection of tasks (see new Figures 11 and 12).
5. Modified the motivation, related work and discussion to better emphasize the focus of our study on optimizing neural network weights, highlight other BBO benchmarks and extend future potential use cases.

For Reviewer ezSc ([link](https://openreview.net/forum?id=s6qtLyR6uJ&noteId=GzAia6yziI)), we

1. Answered all the reviewer’s questions.
2. Reported additional experiments on BBOB and HPO-B tasks and implemented a Batch Bayesian Optimization baseline (see new Figures 11, 12 and [GitHub link](https://github.com/neuroevobench/neuroevobench/blob/main/neuroevobench/blines/bayes_opt_jax.py)).
3. Noted that while additional tasks can always be added to the modular design of the benchmark, one has to watch the computational resources required to make the benchmark useful to the broader community.
4. Provided further clarifying detail on the experiments conducted for Figure 2 and 4, as well as the trade-off between resource allocation and evaluation noise.

For Reviewer 1pB919 ([link](https://openreview.net/forum?id=s6qtLyR6uJ&noteId=0LCPwTwXpt)), we

1. Answered all the reviewer’s questions.
2. Clarified concerns about the chosen color palette and pointed the reviewer to figure 1 with respect to the aggregated benchmark results.
3. Enhanced the discussion of prior work on benchmarking black-box optimizers highlighting that the majority of these are solely focusing on hyperparameter optimization.
4. Updated the wording and title to better reflect our target audience.

For Reviewer geZa ([link](https://openreview.net/forum?id=s6qtLyR6uJ&noteId=tUmRvBSbAp)), we

1. Answered all the reviewer’s questions.
2. Implemented a Batch Bayesian Optimization baseline (see [GitHub link](https://github.com/neuroevobench/neuroevobench/blob/main/neuroevobench/blines/bayes_opt_jax.py)).
3. Fixed a dependency issue in the colab notebook.

We hope to have addressed the reviewer’s concerns. If there remain further questions, we are happy to discuss and address them.

---

### Author Response · Authors · 2023-08-28
**Added Batch Bayesian Optimization results & end of discussion period**

Dear reviewers,

we quickly want to update you on the most recent results we have added during the rebuttal period. All of them can be found in this slide set ([Link](https://docs.google.com/presentation/d/1okxaIes937RO7017bLYUjqhmbakIbqRImwqrTQZUAOs/edit?usp=sharing)). More specifically, we have

1. Added a Batch Bayesian Optimization baseline to the benchmark (see [GitHub link](https://github.com/neuroevobench/neuroevobench/blob/main/neuroevobench/blines/bayes_opt_jax.py)).
2. Added results for Bayesian Optimization including neuroevolution, BBOB and HPO-B tasks (see figures in slide deck).
3. Added a set of visualizations to illuminate the hyperparameter robustness of the evolutionary optimization methods used in the benchmark study (see new figures 13-21).
4. Added a set of experiment results on non-neural network tasks to highlight the need for a separate benchmark tailored to the Deep Learning community and clarified the selection of tasks (see new Figures 11 and 12).

We hope that these results as well as the individual comments and manuscript revision have addressed the concerns of the reviewers. Given that the discussion period is ending soon, we would like to kindly ask the reviewers to provide feedback on our changes and additional work.

Best wishes,

The authors

---

### Decision · Program_Chairs · 2023-09-22

**Decision:**

Accept (Poster)

**Comment:**

Neuroevolution is an interesting and promising research field. Whereas one reviewer's evaluation is negative, I think that this paper is acceptable (based on the other three reviewers' positive evaluations and my own evaluation about the potential future growth of this field). It would be an interesting future research topic to provide a benchmarking framework for multi-objective neuroevolution.